# FGAIF: Aligning Large Vision-Language Models with Fine-grained AI Feedback

**Liqiang Jing**   **Xinya Du**

*Department of Computer Science, The University of Texas at Dallas*
*jingliqiang6@gmail.com, xinya.du@utdallas.edu*

**Reviewed on OpenReview:** *https://openreview.net/forum?id=Qhfw5CUVd7*

## Abstract

Large Vision-Language Models (LVLMs) have demonstrated proficiency in tackling a variety of visual-language tasks. However, current LVLMs suffer from misalignment between text and image modalities which causes three kinds of hallucination problems, i.e., object existence, object attribute, and object relationship. To tackle this issue, existing methods mainly utilize Reinforcement Learning (RL) to align modalities in LVLMs. However, they still suffer from three main limitations: (1) General feedback can not indicate the hallucination type contained in the response; (2) Sparse rewards only give the sequence-level reward for the whole response; and (3) Annotation cost is time-consuming and labor-intensive. To handle these limitations, we propose an innovative method to align modalities in LVLMs through **F**ine-**G**rained **A**rtificial **I**ntelligence **F**eedback (**FGAIF**), which mainly consists of three steps: AI-based Feedback Collection, Fine-grained Reward Model Training, and Reinforcement Learning with Fine-grained Reward. Specifically, We first utilize AI tools to predict the types of hallucination for each segment in the response and obtain a collection of fine-grained feedback. Then, based on the collected reward data, three specialized reward models are trained to produce dense rewards. Finally, a novel fine-grained feedback module is integrated into the Proximal Policy Optimization (PPO) algorithm. Extensive experiments are conducted on hallucination and general benchmarks, demonstrating the superior performance of our proposed method. Notably, compared with previous models trained with the RL-based aligning method, our proposed method is effective even with fewer parameters.

## 1 Introduction

Large Language Models (LLMs) like GPT-3 (Brown et al., 2020) and ChatGPT (OpenAI, 2022) have showcased remarkable abilities in language processing. However, their ability to handle multimodal inputs combining both visual and textual data remains inadequate. This limitation has drawn research attention to Large Vision-Language Models (LVLMs) which achieve massive success in various vision and language tasks (e.g. Visual Question Answering (Antol et al., 2015), Visual Entailment (Zhang et al., 2025), and Image Captioning (Lin et al., 2014)).

Although LVLMs have achieved significant success in tasks requiring visual-textual understandings, the challenge of misalignment between vision and language modalities (Sun et al., 2023) has not been solved, leading to "hallucination" in generated textual responses (Jing et al., 2024). As shown in Figure 1, there are three kinds of hallucinations in the context of LVLMs, including (1) Object Existence Hallucination, where non-existent objects are mistakenly referenced; (2) Object Attribute Hallucination, involving inaccuracies in the depiction of object attributes like color, shape, and size; and (3) Object Relationship Hallucination, where the descriptions inaccurately portray the interactions or spatial relationships between objects, leading to misrepresentations of their positions, interactions, and actions involving two or more objects Jing et al.

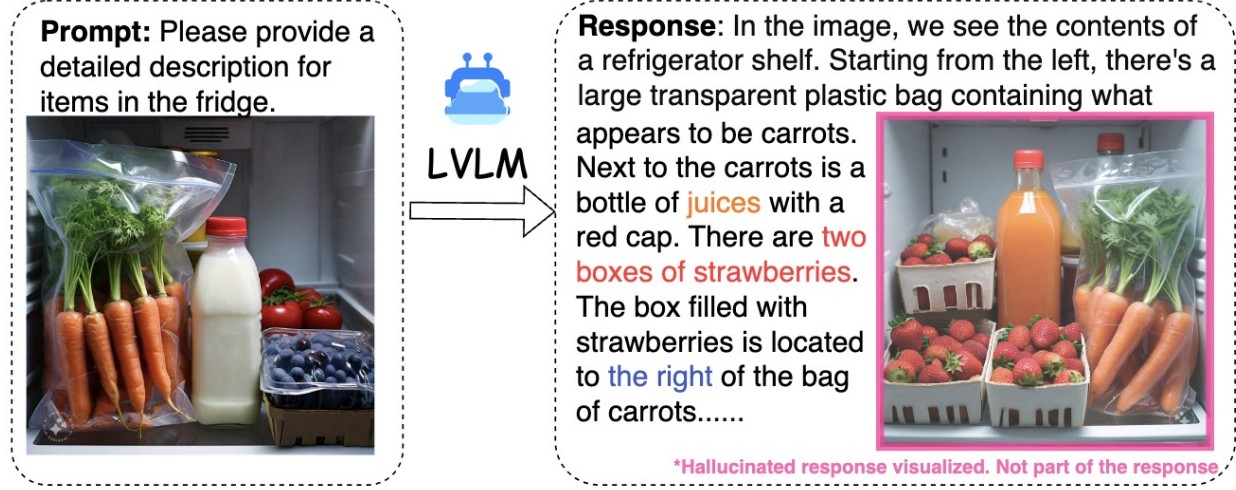

Figure 1: Illustration of the hallucination in the response generated by the LVLM. We illustrate all three kinds of hallucinations in this figure, where orange fonts denote object existence hallucinations, red fonts denote object attribute hallucinations, and blue fonts for object relation hallucinations.

(2024); Zhai et al. (2023); Zhang et al. (2024). Therefore, mitigating the hallucinations and generating faithful responses are key to building practical applications of LVLMs.

Hallucinations in LVLMs stem from their inclination to lean on common sense or stereotypical knowledge ingrained in the textual data used for training and frequently ignore the visual information presented (Cui et al., 2023), where the specific details contained in the input images (Zhou et al., 2024) are greatly overlooked. Such discrepancies are largely caused by the misalignment between textual and visual modalities (i.e., modality misalignment problem). To tackle this kind of misalignment problem, most existing methodologies rely on Reinforcement Learning (RL) (Ziegler et al., 2019; Sun et al., 2023; Li et al., 2023a; Zhou et al., 2024). For example, LLaVA-RLHF (Sun et al., 2023) aims to first gather human preferences and then incorporate these preferences into the reinforcement learning process for fine-tuning Language Models.

Despite their great success, the existing modality alignment method still suffers from three limitations: (1) General Feedback. Only broad and general feedback is generated by the reward model employed in current methodologies, and hallucination of specific types like objects and relations is not contained, making it challenging to precisely identify and correct inaccuracies in the generated content in the training stage. (2) Sparse Rewards. During the modality alignment training process, sequence-level feedback is gathered by current methodologies for the entirety of long responses, which is a kind of sparse training signal and is suitable to the task requiring the generation of long-form text. Moreover, sequence-level feedback tends to overlook the detailed hallucinations that may occur within individual segments of the response. (3) High Annotation Costs. Prevailing methods primarily utilize rewards based on human annotations, which is time-consuming and labor-intensive. Thus, scalability is another constraint for existing methods requiring massive accurate feedback.

To mitigate above-mentioned limitations, we propose to align modalities in large vision-language models with Fine-Grained AI Feedback (FGAIF), an innovative approach to refine large vision-language models via fine-tuning. In particular, our method mainly consists of three steps: AI-based feedback collection, fine-grained reward model training, and reinforcement learning with fine-grained rewards. The AI-based feedback collection step provides three kinds of segment-level (i.e., sub-sentence-level) hallucination labels based on AI feedback. We train three reward models that can produce fine-grained rewards, i.e., multiple types and segment-level rewards, using the collected fine-grained reward data, in the second step. The last step integrates novel fine-grained feedback into the Proximal Policy Optimization (PPO) algorithm to further fine-tune the LVLM.

Our contribution can be summarized as follows:

- We propose a novel fine-grained artificial intelligence-based hallucination labeling method, which can detect three types of hallucinations (i.e., object existence, object attribute, and object relation) in terms of sub-sentence level and eliminate the need for manual annotation.

- To the best of our knowledge, we are the first to provide multiple types and segment-level feedback towards modalities alignment in LVLMs, which can mitigate three kinds of hallucination in LVLMs.

- We conduct comprehensive experiments on several hallucination benchmarks and one general benchmark[1]. The experimental results demonstrate the effectiveness of FGAIF. In addition, the ablation study shows the necessity of each module in FGAIF.

## 2 Related Work

### 2.1 Large Vision-Language Model

The recent pivot of the multimodal learning community towards LVLMs has been largely inspired by the effective pretraining approaches seen in LLMs and Vision Foundation Models (VFMs). At the heart of modern advanced LVLMs lie three fundamental components: a text encoder, an image encoder, and a cross-modal alignment module (Rohrbach et al., 2018). The text encoder typically manifests as a language model, with notable examples being LLaMA (Touvron et al., 2023) and Vicuna (Chiang et al., 2023), whereas the image encoder usually borrows from VFMs like ViT (Dosovitskiy et al., 2021). The critical role of the cross-modal alignment module is to fuse the visual and textual domains, thereby enabling the text encoder to grasp visual semantics more effectively. LVLMs generally undergo a multi-stage training approach to master visual comprehension (Gong et al., 2023; Zhu et al., 2023; Liu et al., 2023b;c; Ye et al., 2023; Dai et al., 2023). For example, Liu et al. (2023c) initially pre-trains the model by aligning image features with the word embeddings from a pre-trained LLM, followed by fine-tuning on specific language-image instruction datasets. To boost training efficiency, LVLMs often employ techniques like freezing parameters in the LLM or VFM components and utilize efficient fine-tuning methods such as LoRA (Hu et al., 2022b).

Despite their significant progress, LVLMs still face challenges with hallucinations, which can severely affect their performance on various vision-language tasks (Rohrbach et al., 2018).

### 2.2 Hallucinations in LVLMs

Motivated the hallucination in LLMs, more researchers shifted research attention to hallucination in LVLMs. Hallucination in the context of LVLMs is the inconsistent content between the generated response and the input image. To evaluate the hallucination in LVLMs, some work devised metrics to measure the hallucination in the response, such as FaithScore (Jing et al., 2024), CHAIR (Rohrbach et al., 2018), POPE (Li et al., 2023d), and NOPE (Lovenia et al., 2023). Recently, there have been works to mitigate hallucinations in LVLMs utilizing various technologies, such as decoding approaches (Leng et al., 2023; Huang et al., 2023), post-processing (Zhou et al., 2023a; Yin et al., 2023; Chang et al., 2024), and construction of the higher-quality dataset (Liu et al., 2023a; Li et al., 2023c). To address the challenge of aligning image and text modalities within LVLMs and to mitigate the issue of hallucination, existing strategies offer partial solutions but lack direct guidance for modality alignment. Therefore, some research efforts (Li et al., 2023b; Yu et al., 2023; Zhou et al., 2024) have embraced the use of reinforcement learning for direct modality alignment. For example, Sun et al. (2023) developed the LLaVA-RLHF model, harnessing human-annotated preference data to minimize hallucinations in LLaVA.

Motivated by the fine-grained RL (Wu et al., 2023; Ramé et al., 2023; Jang et al., 2023; Zhou et al., 2023b; Wang et al., 2024) and AI-based RL (Lee et al., 2023; Bai et al., 2022) methods, we propose to align modalities in LVLMs with fine-grained AI feedback. Different from existing work which needs human annotation and only provides coarse-grained feedback, our method provides fine-grained rewards and learns from AI automatic feedback.

---

[1]We released our code via `https://github.com/LiqiangJing/FGAIF`.

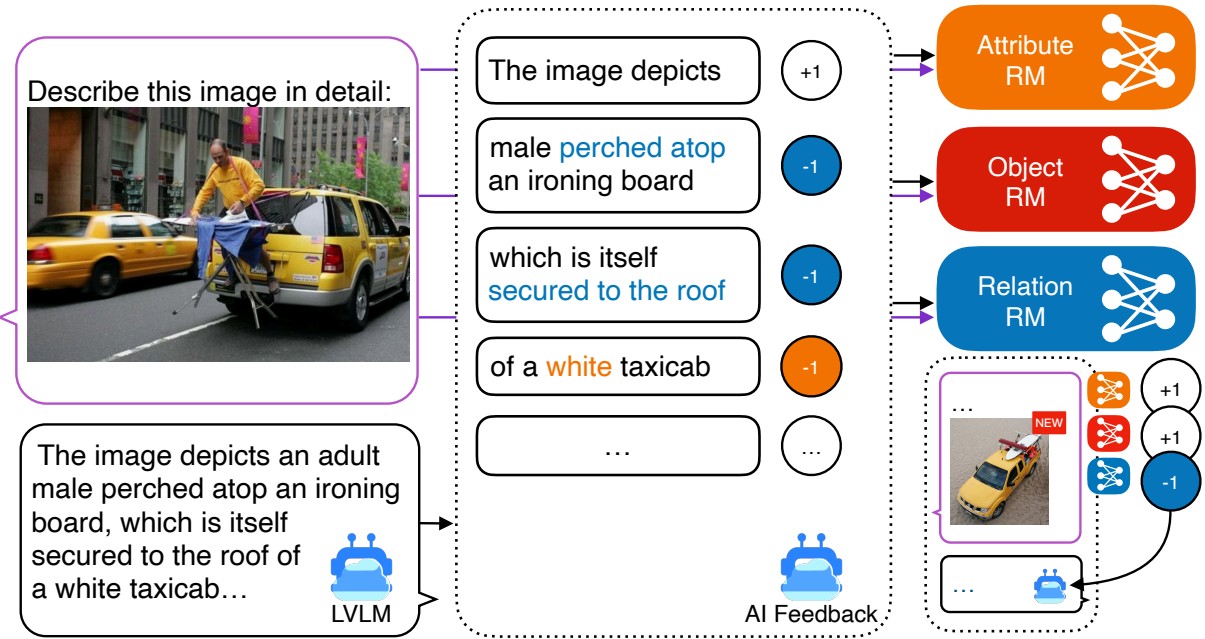

Figure 2: The illustration of our proposed FGAIF, which consists of three steps: AI-based feedback collection, fine-grained reward model training, and reinforcement learning with fine-grained rewards.

## 3 Problem Formulation

Suppose we have a set of $N$ images $\{I_i\}_{i=1}^N$ and the corresponding prompts $\{P_i\}_{i=1}^N$. Next, we omit the index of $I_i$ and $P_i$ for simplicity. Then we feed the prompt $P$ and image $I$ into an LVLM $\mathcal{M}$ and get the sampled response as $R = \mathcal{M}(I, P|\Theta_M)$, where $R$ is the response for $(I, P)$. $\Theta_M$ refers to the parameters of LVLM $\mathcal{M}$. Next, we resort to another AI-based method $\mathcal{A}$ to identify three kinds of hallucination (i.e., object existence, object attribute, and object relation ) in the generated response and **train three reward models** as $F^o, F^a, F^r = \mathcal{A}(R, I, P), \mathcal{R}^{o/a/r}(R, I, P|\Theta_{o/a/r}) \to F^{o/a/r}$, where $F^{o/a/r} = \{f_1^{o/a/r}, \cdots, f_s^{o/a/r}\}$ denotes the object existence/attribute/relation hallucination labels. $\Theta_{o/a/r}$ is the parameters of the reward model $\mathcal{R}^{o/a/r}$. $f_j^{o/a/r}$ is the label which means whether the $j$-th sub-sentence in the response contains the object existence/attribute/relation hallucination. $\mathcal{R}^{o/a/r}$ denotes reward models which aim to detect object existence/attribute/relation hallucinations.

Finally, we utilize well-trained reward models and the images $I^f$ and the corresponding prompt $P^f$. to **fine-tune the LVLM** to make it generate faithful responses as $\hat{R} = \mathcal{M}(I^f, P^f, |\Theta_f, \mathcal{R}^o, \mathcal{R}^a, \mathcal{R}^r)$, where $\Theta_f$ is final optimized parameters of the LVLM $\mathcal{M}$. We also omit the index in this equation.

## 4 Methodology

In this section, we detail the proposed FGAIF, which consists of three steps: AI-based feedback collection, fine-grained reward model training, and reinforcement learning with fine-grained rewards.

### 4.1 AI-based Feedback Collection

In our method, we explore a reward function informed by multiple detailed reward models for aligning modalities in LVLMs. These models (1) provide rewards at frequent intervals (namely, for sub-sentence of the generated content) and (2) assign rewards according to various categories of hallucinations. Each

category of hallucination is evaluated by a distinct reward model. Therefore, in this stage, to train the reward model that can detect the hallucination, we collect the reward dataset first. Different from the most existing work which collects coarse-grained reward data via human feedback to refine VLMs, we collect fine-grained reward data by automatic AI model (left of Figure 2).

To achieve this, we first sample responses from the backbone LVLM as depicted in Section 3. Inspired by the existing fine-grained evaluation work (Jing et al., 2024; Min et al., 2023), we devise a fine-grained AI-based feedback collection method. In particular, we utilize AI models to annotate three kinds of hallucinations (i.e., object existence hallucination, object attribute hallucination, and object relationship hallucination) on the sub-sentence level for the response. In particular, to get the hallucination labels for each sub-sentence, we first split the response from the LVLM into sub-sentences as follows,

$$(s_1, \cdots, s_n) = \text{SPLIT}(R), \tag{1}$$

where $s_i$ is the $i$-th sub-sentence of the response. Thereafter, to accurately annotate three kinds of hallucination in the sub-sentence, we extract three kinds of atomic facts (Jing et al., 2024): object existence, object attribute, and object relationship atomic facts, from the sub-sentence, using ChatGPT as follows,

$$\{\{a_1^o, \cdots, a_i^o, \cdots\}, \{a_1^a, \cdots, a_i^a, \cdots\}, \{a_1^r, \cdots, a_i^r, \cdots\}\} \tag{2}$$
$$= \text{ChatGPT}(\text{P}_s(s, \{s_i\}_{i=1}^n)),$$

where $a_i^o$, $a_i^a$ and $a_i^r$ denote the $i$-th object existence, object attribute, and object relation types of atomic fact derived from the sub-sentence, respectively. $\text{P}_s(\cdot)$ is a prompt that can instruct ChatGPT to generate three kinds of atomic facts, as shown in Figure 3.

Thereafter, to get the label of each type of hallucination for each sub-sentence, we need to verify whether the atomic fact is consistent with the input image. We utilize superior LLaVA 1.5 (Liu et al., 2023b) to annotate the object existence hallucination, attribute hallucination, and relationship hallucination. Specifically, we feed LLaVA 1.5 with the image, the atomic fact, and the prompt, which can instruct LLaVA 1.5 to identify the consistency between atomic facts and the input image as follows,

$$f_{a_i}^{o/a/r} = LLaVA(\text{P}_{con}(I, a_i^{o/a/r})), \tag{3}$$

where $f_{a_i}^o \in \{0, 1\}$, $f_{a_i}^a \in \{0, 1\}$ and $f_{a_i}^r \in \{0, 1\}$ denote the hallucination label of $i$-th atomic fact in the sub-sentence in terms of object existence, object attribute, and object relationship types of atomic facts, respectively. $f_{a_i}^{o/a/r}$ is set to 1 when the output of LLaVA 1.5 indicates that the input image and the atomic fact are inconsistent (i.e., the corresponding atomic fact is a hallucination), otherwise, it is set to 0. $\text{P}_{con}(\cdot)$ is the prompt that can be used to prompt the LLaVA 1.5 to annotate hallucination and it is shown in Figure 4.

Finally, we can aggregate the hallucination labels of atomic facts for each sub-sentence and then get the fine-grained sub-sentence-level hallucination labels as $f^{o/a/r} = sgn(\sum_i f_{a_i}^{o/a/r})$, where $f^{o/a/r}$ is the hallucination label for the sub-sentence in terms of object existence/attribute/relation. $sgn(\cdot)$ is the sign function. In addition, if there is not any atomic fact in a sub-sentence, the corresponding label $f^{o/a/r}$ is set to 2.

The reason why we use LVLM to verify the consistency between atomic fact and image even if the LVLM may also introduce hallucination: Our method converts the AI labeling task into a discriminative task that usually generates a short response, and this kind of task tends not to generate hallucination, which has been demonstrated in existing work (Jing et al., 2024; Min et al., 2023). Therefore, our AI-based feedback collection method can reduce the hallucination as much as possible.

## 4.2 Fine-grained Reward Model Training

As mentioned before, the existing LVLMs mainly suffer from three aspects of hallucinations, i.e., object existence, object attribute, and object relation. Based on the process above, we can get three kinds of hallucination labels for each sample. Thereafter, we train three reward models corresponding to each kind of

Given an answer output by a vision-language model, break down its sub-sentence into independent atomic facts from it.
First extract elements from the answer. Then classify each element into a category (object, attribute, relation).
Finally, generate atomic facts for each element. You can refer to the context of the sub-sentence.
The relation must be the relationship between two objects.
Please note that you only need to output atomic facts. Besides, you must follow the format of examples. Facts are separated directly by periods.
The context is: %s
Please do not output other irrelevant information.

You should convert the pronoun into a specific object according to the context.
Please note that you only need to output atomic facts that are in the sub-sentence, the context is only used to help you understand context information such as the object to which the pronoun refers, don't output any content that didn't appear in the given sub-sentence.
Please note that the object is an objective description, not a subjective analysis, such as the atmosphere is not an object.
If the sub-sentence does not contain any object/attribute/relation, leave the corresponding line empty such as Object:

Sub-sentence: A man posing for a selfie in a jacket and bow tie.
Atomic facts:
Object: There is a man. There is a selfie. There is a jacket. There is a bow tie.
Attribute:
Relation: A man is in a jacket. A man is in a bow tie. A man posing for a selfie.

Sub-sentence: The image features a red velvet couch with a cat lying on it.
Atomic facts:
Object: There is a couch. There is a cat.
Attribute: The couch is red. The couch is velvet.
Relation: A cat is lying on a couch.

Sub-sentence: The photo is about a close-up image of a giraffe's head.
Atomic facts:
Object: There is a giraffe's head.
Attribute:
Relation:

Sub-sentence: A horse and several cows feed on hay.
Atomic facts:
Object: There is a horse. There are cows. There is a hay.
Attribute:
Relation: A horse feeds on hay. Cows feed on hay.

Sub-sentence: A red colored dog.
Atomic facts:
Object: There is a dog.
Attribute: The dog is red.
Relation:

Sub-sentence: {sub-sentence}
Atomic facts:

Figure 3: The prompt of atomic fact generation. In this prompt, we asked ChatGPT to generate three kinds of atomic facts: object existence, object attribute, and object relation. To get better performance on atomic fact generation, we added some samples in this prompt.

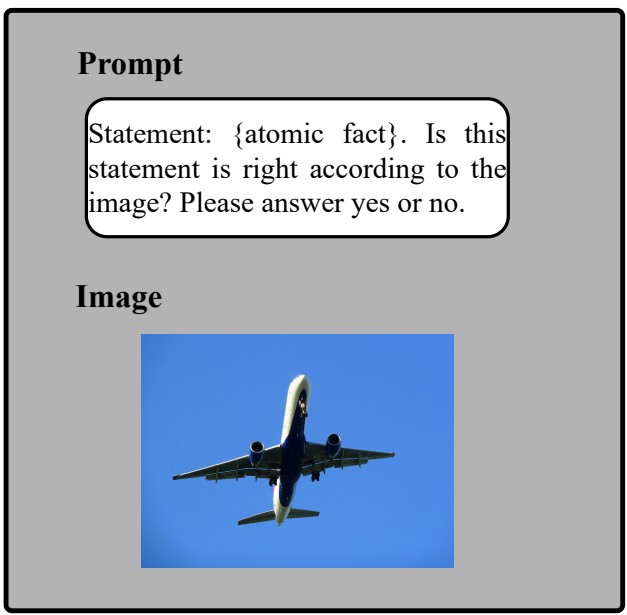

**Prompt**

Statement: {atomic fact}. Is this statement is right according to the image? Please answer yes or no.

**Image**

Figure 4: The prompt for verifying the consistency between the image and atomic fact.

hallucination (middle of Figure 2). Specifically, we first split the input of the reward model into tokens and get the index of the last token of each sub-sentence for the subsequent hallucination prediction as follows,

$$\begin{cases} T = \text{Tokenizer}([P, I, R]), \\ \{ind_1, \cdots, ind_n\} = \text{Search}([P, I, R, T]), \end{cases} \tag{4}$$

where $ind_i$ is the index of the last token of the $i$-th sub-sentence. $n$ is the total number of sub-sentences and $T$ is the tokens for the input $R$ (response), $P$ (prompt) and $I$ (image). Seach is a function that can get the index of the last token for each sub-sentence.

Finally, we can utilize the above-recognized indices to train reward models which is able to detect various kinds of hallucinations in the sub-sentence of response. In particular, we first feed the tokens above into the reward model backbones as follows,

$$\mathbf{F}^o = \text{RM}^o(T), \mathbf{F}^a = \text{RM}^a(T), \mathbf{F}^r = \text{RM}^r(T). \tag{5}$$

Then, we connect the output from reward models, corresponding to the last token, with an MLP classifier. Thereafter, we can predict the hallucination label with the classifier. The above process can be formulated as follows,

$$\hat{f}_j^{o/a/r} = \text{MLP}_{o/a/r}(\mathbf{F}_{ind_j}^{o/a/r}), \tag{6}$$

where $\mathbf{F}_{ind_j}^{o/a/r}$ is the feature vector of the last token for the $j$-th sub-sentence. $\hat{f}_j^o$, $\hat{f}_j^a$ and $\hat{f}_j^r$ are the predicted labels. To equip the three reward models with hallucination detection ability and give further rewards for reinforcement learning, we train the three reward models with a cross-entropy loss as $\mathcal{L}_{o/a/r} = \sum_{j=1}^n CE(f_j^{o/a/r}, \hat{f}_j^{o/a/r})/n$, where $CE(\cdot)$ is the cross-entropy function and $\mathcal{L}_o$, $\mathcal{L}_a$ and $\mathcal{L}_r$ are loss functions for different reward models (i.e., object existence, object attribute, and object relation).

### 4.3 Reinforcement Learning with Fine-grained Reward

Fine-tuning language models with reinforcement learning is an effective approach to align modalities in LVLMs. To make LVLMs generate more faithful responses rather than hallucinated responses, we also resort to reinforcement learning to further fine-tune LVLMs with the fine-grained reward (right of Figure 2).

Specifically, we first segment the generated response from the LVLM into $K$ sub-sentences $(s^1, \cdots, s^K)$. Then we get all kinds of rewards for each sub-sentence based on the well-trained reward model by cross-entropy loss. We define $r_o^i$, $r_a^i$, and $r_r^i$ as the object existence, object attribute, and object relation rewards for the $j$-th sub-sentence. Then we have a combined reward function for each token as $r_t = \sum_{l \in \{o,a,r\}} \sum_{i=1}^{K} (\mathbb{I}(t = T_i) w_l r_l^i)$, where $T_i$ is the timestep for the last token of $s^i$. $\mathbb{I}(\cdot)$ is the indicator function. $w_l \in \mathbb{R}$ is a weight assigned to rewards. Thereafter, we utilize the PPO algorithm Schulman et al. (2017) to train the policy model (i.e., the LVLM) following the existing work (Sun et al., 2023).

PPO is an actor-critic RL algorithm that is widely used in previous RLHF work to optimize the policy model against a reward model of human feedback. It uses a value model $V_\psi(s_t)$ to estimate the value of state $s_t$, and optimizes the policy model with a PPO clipped surrogate training objective. The advantage $A_t$ at timestep $t$ is estimated by a generalized advantage estimation function (Schulman et al., 2016): $A_t = \sum_{t'=t}^{T} (\gamma\lambda)^{t'-t}(r_{t'} + \gamma V_\psi(s_{t'+1}) - V_\psi(s_{t'}))$, with $\gamma$ as a hyperparameter and $\lambda$ as the discounting factor for rewards. $r_t$ is the reward assigned to $a_t$, which in our case is acquired using one or multiple learned reward models. The value model $V_\psi(s_t)$ is optimized with an expected squared-error loss with the value target as $V_{\text{targ}}(s_t) = \sum_{t'=t}^{T-1} \gamma^{t'-t} r_{t'} + \gamma^{T-t} V_{\psi_{\text{old}}}(s_T)$, where $V_{\psi_{\text{old}}}$ is the lagging value model. Finally, PPO is trained to optimize both policy ($P_\theta$) and value ($V_\psi$) models with their respective objectives. No reward model is being optimized during PPO training. See Appendix C for more details.

## 5 Experiment

In this section, we present the extensive experiments to answer the following research questions: 1) **RQ1.** What is the quantitative performance of our FGAIF? 2) **RQ2.** What is the contribution of each component of FGAIF? 3) **RQ3.** What is the intuitive performance of our FGAIF?

Table 1: POPE evaluation benchmark. Accuracy denotes the accuracy of predictions. "Yes" represents the probability of the model outputting a positive answer. ↑ denotes that the larger the value, the better the performance. The bold font denotes the best performance among our model and baselines with the same backbone architecture (LLaVA). The underlined font denotes the second-best performance among our model and baselines with the same backbone architecture.

| Model | POPE | | | | | | | |
| | Random | | Popular | | Adversarial | | Overall | |
| | Acc↑ | F1↑ | Acc↑ | F1↑ | Acc↑ | F1↑ | F1↑ | Yes |
|---|---|---|---|---|---|---|---|---|
| MiniGPT-4$_{7B}$ | 79.7 | 80.2 | 69.7 | 73.0 | 65.2 | 70.4 | 74.5 | 60.8 |
| mPLUG-Owl$_{7B}$ | 54.0 | 68.4 | 50.9 | 66.9 | 50.7 | 66.8 | 67.2 | 97.6 |
| InstructBLIP$_{7B}$ | 88.6 | 89.3 | 79.7 | 80.2 | 65.2 | 70.4 | 80.0 | 59.0 |
| InstructBLIP$_{13B}$ | 88.7 | 89.3 | 81.4 | 83.5 | 74.4 | 78.5 | 83.7 | 62.2 |
| LLaVA$_{7B}$ | 50.4 | 66.6 | 49.9 | 66.4 | 49.7 | 66.3 | 66.4 | 99.2 |
| LLaVA$_{13B}$ | 73.7 | 78.8 | 73.6 | 78.2 | 67.2 | 74.4 | 77.1 | 73.7 |
| LLaVA-RLHF$_{7B}$ | 84.8 | 83.3 | 83.3 | 81.8 | 80.7 | 79.5 | 81.5 | 41.8 |
| LLaVA-RLHF$_{13B}$ | 85.2 | 83.5 | 83.9 | 81.8 | **82.3** | **80.5** | 81.9 | 39.0 |
| FGAIF$_{7B}$ | **87.0** | **86.7** | **84.0** | **83.7** | 79.6 | 79.9 | **83.4** | 48.3 |

### 5.1 Experimental Details

To ensure a fair and equitable comparison, we utilized same base model with the LLaVA-RLHF model whose network architecture is LLaVA$_{7B}$ and underwent a supervised fine-tuning stage. In addition, we also adopt the same architecture (i.e., LLaVA$_{13B}$) with LLaVA-RLHF for the reward model. We compared our method with these models that used the same model backbone as ours (i.e., **LLaVA**$_{7B}$ (Liu et al., 2023c) and **LLaVA-RLHF**$_{7B}$). We also introduced some methods with the same backbone architecture but a larger model size (i.e., **LLaVA**$_{13B}$ and **LLaVA-RLHF**$_{13B}$). Besides, we further incorporated more advanced LVLMs for comparison, i.e., MiniGPT-4$_{7B}$ (Zhu et al., 2023), mPLUG-Owl$_{7B}$ (Ye et al., 2023),

Table 2: Evaluation results for different LLMs on MMHal-Bench and LLaVA-Bench. "Over" and "Hal" denotes "Overall Score" and "Hallucination Rate", respectively. "Con", "De" and "Com" denote "Conversation", "Detailed Description", and "Complex Question".

| Model | MMHal-Bench | | | | | LLaVA-Bench | | | |
| | Over↑ | Hal ↓ | Object↑ | Attribute↑ | Relation↑ | Con↑ | De↑ | Com↑ | Full↑ |
|---|---|---|---|---|---|---|---|---|---|
| MiniGPT-4$_{7B}$ | 3.39 | 0.24 | 3.0 | 2.54 | 3.67 | 80.5 | 74.5 | 81.6 | 78.9 |
| mPLUG-Owl$_{7B}$ | 2.49 | 0.43 | 0.33 | 2.58 | 1.5 | 78.7 | 46.0 | 47.4 | 57.5 |
| InstructBLIP$_{7B}$ | 2.10 | 0.58 | 2.08 | 2.67 | 2.17 | 95.4 | 96.3 | 99.1 | 97.0 |
| InstructBLIP$_{13B}$ | 2.14 | 0.58 | 1.75 | 2.82 | 2.5 | 90.9 | 91.7 | 109.3 | 97.2 |
| LLaVA$_{7B}$ | 1.55 | 0.76 | 0.00 | 1.25 | 2.00 | 75.1 | 75.4 | 92.3 | 81.0 |
| LLaVA$_{13B}$ | 1.11 | 0.84 | 0.00 | 1.13 | 1.5 | 87.2 | 74.3 | 92.9 | 84.9 |
| LLaVA-RLHF$_{7B}$ | 2.04 | 0.68 | 1.83 | 2.42 | 2.25 | 93.0 | 79.0 | 109.5 | 94.1 |
| LLaVA-RLHF$_{13B}$ | _2.53_ | _0.57_ | _2.67_ | _2.79_ | _2.33_ | _93.9_ | _82.5_ | **110.1** | _95.6_ |
| FGAIF$_{7B}$ | **3.09** | **0.36** | **3.58** | **3.21** | **3.33** | **98.2** | **93.6** | _110.0_ | **100.1** |

InstructBLIP$_{7B}$ (Dai et al., 2023), and InstructBLIP$_{13B}$. The size of LLaVA in Equation (2) is 13B and ChatGPT in Equation (3) is gpt-3.5-turbo-1106.

To verify the effectiveness of our proposed FGAIF, we compare our method with baselines on several benchmarks, including **QA-based hallucination benchmarks** POPE (Li et al., 2023d) and MMHal-Bench (Sun et al., 2023), **hallucination metrics** CHAIR (Rohrbach et al., 2018) and FaithScore (Jing et al., 2024), and the **general** benchmark LLaVA-Bench (Liu et al., 2023c).

**POPE** is a framework specifically designed for assessing object existence hallucinations in LVLMs. Specifically, POPE formulates the evaluation of object hallucination as a binary classification task that prompts LVLMs to output "Yes" or "No", e.g., "Is there a chair in the image?" "Yes" questions can be directly constructed based on objects appearing in the image. The "No" questions are constructed by three distinct sampling settings: random, popular, and adversarial. In the random setting, objects that are not present in the image are selected randomly. For the popular setting, the chosen non-existent objects are those from a pool of objects that appear most frequently in the MSCOCO dataset. In the adversarial setting, the sampling negative objects are often seen together with the objects in the image but are absent in the image under evaluation. This comprehensive approach allows for a nuanced analysis of the model's tendency to hallucinate across different scenarios. Finally, POPE consists of 3,000 samples under the setting of each type of negative sampling and 9,000 samples for the whole dataset.

**MMHal-Bench** benchmark has been introduced to assess and measure the degree of hallucination in responses by LVLMs. MMHAL-BENCH comprises 96 carefully constructed image-question pairs across eight different question categories and 12 object topics. These pairs are crafted to challenge LVLMs on common points of failure, including 1) Object Attribute, 2) Adversarial Object, 3) Comparison, 4) Counting, 5) Spatial Relation, 6) Environment, 7) Holistic Description, 8) Others. Different with POPE, it can evaluate more fine-grained hallucinations rather than only object existence.

**CHAIR** is a framework to quantify object hallucination in image captions. This method compares objects generated in captions against the ground truth objects within the images. CHAIR assesses hallucination on two levels: sentence-level and instance-level. The sentence-level score, referred to as CHAIR$_S$, quantifies the proportion of captions that contain hallucinated content, whereas the instance-level score, CHAIR$_I$, measures the frequency of hallucinated objects relative to the total number of objects mentioned by the model. Our evaluation involves a randomly selected subset of 1,000 images from the MSCOCO validation set, allowing for an analysis of our model's performance in minimizing object existence hallucination.

**FaithScore** is another framework to assess the accuracy and relevance of response generated by LVLMs. This innovative approach focuses on evaluating the consistency of atomic facts within the response against the depicted scenes in the input images. Different from CHAIR, FaithScore can demonstrate the model's

hallucination performance in terms of object existence, attribute, and relation. Our evaluation involves a randomly selected subset of 1,000 images from the MSCOCO validation set, allowing for an analysis of our model's performance in mitigating object existence, attribute, and relation hallucination. It also provides an instance-level score F-Score and sentence-level score F-Score$_S$.

**LLaVA-Bench** is a general benchmark to assess the performance of LVLMs. LLaVA-Bench consists of 90 samples which can be categorized into three categories: detailed description, conversation, and complex question. All the prompts in this benchmark and answers are generated by GPT-4. In the evaluation process, the standard answer and generated response are fed into GPT-4 and GPT-4 then given a rating. Following the existing work (Sun et al., 2023), we also report the relative scores of LVLMs compared to GPT-4.

All experiments are conducted on a $4 \times$ A100 80G GPU Server. For the reward model training, we use the Adam optimizer, and the learning rate, batch size, and epoch are set to 2e-5, 4, and 100. For the PPO training, we use the Adam optimizer, and the learning rate, batch size, and epoch are set to 1e-7, 256, and 2. We sample 3,500 and 14,000 examples from the MSCOCO 2014 (Lin et al., 2014) training set for reward model training and LVLM training, respectively. The prompt is set to "Describe this image in detail." for model training and sample. we adopt LoRA Hu et al. (2022a) for all the reward model training and the LVLM fine-tuning processes.

Table 3: Results of CHAIR and FaithScore on LVLMs.

| Model | CHAIR | | FaithScore | | Length |
|---|---|---|---|---|---|
| | CHAIR$_I\downarrow$ | CHAIR$_S\downarrow$ | F-Score $\uparrow$ | F-Score$_S\uparrow$ | |
| MiniGPT-4$_{7B}$ | 9.4 | 17.4 | 63.9 | 61.8 | 245.1 |
| mPLUG-Owl$_{7B}$ | 6.2 | 9.5 | 85.6 | 65.7 | 75.2 |
| InstructBLIP$_{7B}$ | 2.4 | 3.8 | 93.6 | 80.0 | 45.6 |
| InstructBLIP$_{13B}$ | 2.7 | 4.0 | 94.1 | 80.8 | 46.3 |
| LLaVA$_{7B}$ | 9.1 | 22.0 | 88.9 | 72.3 | 216.0 |
| LLaVA$_{13B}$ | 10.3 | 19.8 | 87.9 | 68.3 | 121.0 |
| LLaVA-RLHF$_{7B}$ | 4.6 | 7.0 | 89.3 | 71.1 | 58.8 |
| LLaVA-RLHF$_{13B}$ | 7.7 | 20.3 | 89.7 | 73.8 | 413.8 |
| FGAIF$_{7B}$ | **3.9** | **6.2** | **91.2** | **74.7** | 60.2 |

## 5.2 On Model Comparison (RQ1)

The results on **QA-based hallucination benchmarks** (i.e., POPE and MMHal-Bench) are summarized in Table 1 and Table 2. From this table, we have several observations. (1) LLaVA$_{7B}$ and InstructBLIP$_{7B}$ performs worse than LLaVA$_{13B}$ and InstructBLIP$_{13B}$ on most cases, respectively. Compared with LLaVA$_{13B}$, LLaVA$_{7B}$ has a strong hallucination problem, especially its over-confident problem on POPE. This indicates the importance of model size. (2) LLaVA-RLHF$_{7B}$ is better than LLaVA$_{7B}$, which indicates the superiority of further fine-tuning with human feedback. Notably, LLaVA-RLHF$_{7B}$ even has a better performance compared to LLaVA$_{13B}$, even though the latter has specifically more parameters. (3) Our model consistently performs better than the previous advanced in terms of most metrics and testing sets. This verifies that fine-grained artificial intelligence feedback also can be beneficial for hallucination mitigation in LVLMs. (4) Our FGAIF surpasses LLaVA-RLHF$_{7B}$ across all metrics. This implies the advantage of fine-grained artificial intelligence feedback compared to human feedback. (5) To further understand the performance of our FGAIF, we split the MMHal-Bench into three classes based on the original dataset: a) object existence (class "adversarial object"), b) object attribute (classes "object attribute" and "counting"), and c) object relation (class "spatial relation"). We observe that our method consistently achieves the best performance across all question categories.

We further show the performance of our FGAIF and baselines on **hallucination metrics** CHAIR and FaithScore in Table 3. InstructBLIP$_{7B}$ and InstructBLIP$_{13B}$ achieve the best performance in CHAIR and FaithScore metrics. The potential reason is that these two models tend to generate short answers and these

Table 4: Ablation study of our FGAIF. The best results are highlighted in boldface. "Over" and "Hal" denotes "Overall Score" and "Hallucination Rate", respectively.

| Model | CHAIR | | FaithScore | | POPE | MMHal-Bench | |
|---|---|---|---|---|---|---|---|
| | $CHAIR_I$ ↓ | $CHAIR_S$ ↓ | F-Score ↑ | $F\text{-Score}_S$ ↑ | F1 ↑ | Over ↑ | Hal ↓ |
| $FGAIF_{7B}$ | **3.9** | **6.2** | **91.2** | **74.7** | **83.4** | **3.09** | **0.36** |
| w/o-Obj | 4.7 | 6.8 | 89.9 | 73.1 | 81.5 | 2.31 | 0.56 |
| w/o-Att | 4.1 | 6.3 | 90.3 | 73.7 | 82.4 | 2.56 | 0.45 |
| w/o-Rel | 4.2 | 6.4 | 90.3 | 73.4 | 82.6 | 2.64 | 0.44 |
| w/o-AIF | 4.8 | 7.0 | 89.1 | 72.8 | 81.0 | 1.76 | 0.67 |
| w-Coarse | 4.7 | 7.0 | 89.5 | 72.1 | 81.4 | 2.41 | 0.60 |

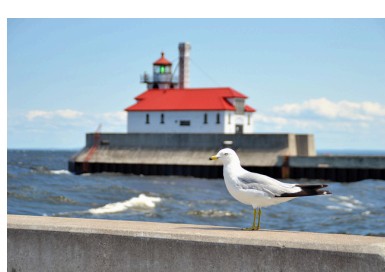

**Prompt**: Generate a short caption of this image.

$LLaVA_{13B}$: A seagull stands on a pier, looking out at a lighthouse and a boat on the water, enjoying the beautiful day by the lake.

Ours: A seagull stands on a concrete ledge near a lighthouse.

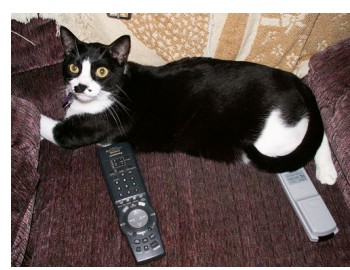

**Prompt**: Generate a short caption of this image.

$LLaVA_{13B}$: A cute black and white cat is lounging on a couch, with a remote control in its mouth, while a television is nearby.

Ours: A black and white cat is laying on a couch next to a remote control.

Figure 5: Comparison between the response generated by our method FGAIF and the baseline $LLaVA_{13B}$ on two testing samples. The red fonts denote the generated hallucinations.

two metrics just measure the precision of faithfulness but do not contain recall of faithfulness. Despite this, our FGAIF still outperforms the RLHF-based baseline (i.e., $LLaVA\text{-}RLHF_{7B}$) whose answers are shorter than FGAIF, which verifies the superiority of our method.

In addition, Table 2 shows the comprehensive performance comparison of our FGAIF and the baseline methods on the **general benchmark** LLaVA-Bench. From this table, we observed that most models perform worst on the "Detail" (i.e., detailed description) subset and perform best on the "Complex" (i.e., complex questions) subset. This may be due to the reason that the "Detail" (i.e., detailed description) subset has more stringent requirements for faithfulness because all the content of the response is required to be an accurate description of the input image. On the contrary, the "Complex" (i.e., complex questions) subset often explores the extended content of an image, sometimes leading to open-ended discussions. Therefore, the demand for strict consistency with the image isn't as critical. In addition, we found that the RLHF can boost the LVLM's performance on the whole LLaVA-Bench from 81.0 ($LLaVA_{7B}$) to 94.1 (LLaVA-$RLHF_{7B}$). Furthermore, our FGAIF can bring more performance gain in terms of the "Conv" subset, "Detail", "Complex" subset, and full set), compared with LLaVA-$RLHF_{7B}$. This further indicates the advance of our method.

### 5.3 On Ablation Study (RQ2)

To verify the effect of each component in our FGAIF, we devise the following variant methods for ablation study:

- **w/o-Obj**: To demonstrate the effect of the object hallucination feedback, we remove the object existence reward model in this method;

- **w/o-Att**: To show the necessity of the attribute hallucination feedback, we remove the object attribute reward model in this method;

- **w/o-Rel**: To demonstrate the effect of the relation hallucination feedback, we remove the object relation reward model in this method;

- **w/o-AIF**: To show the benefit of using reinforcement learning from fine-grained artificial intelligence feedback, we remove all the reinforcement learning components in this variant;

- **w-Coarse**: To verify the advance of the fine-grained feedback compared with the traditional coarse-grained uni reward model, we replace the three fine-grained reward models with one reward model which also is trained with AI annotated data and the training phase is the same as the previous work (Sun et al., 2023).

Table 4 shows the ablation study results of our FGAIF on several hallucination benchmarks. From this table, we have the following observations. 1) w/o-RLAIF performs terribly compared with FGAIF. It confirms the necessity of using RLAIF for modality alignment and hallucination mitigation in LVLMs. 2) FGAIF consistently outperforms w/o-Obj, w/o-Att, and w/o-Rel, across different evaluation metrics. This is reasonable because each reward model can provide feedback for one kind of hallucination. 3) FGAIF surpasses w-Coarse, denoting that the fine-grained reward models are more essential to align modalities in LVLMs compared with the traditional coarse-grained uni reward model. 4) w/o-Obj performs worse than w/o-Att and w/o-Rel. This indicates that the object existence hallucination is most important. The potential reason is that there are more atomic facts of object existence, compared to the other hallucinations. 5) w/o-Att show a similar performance with w/o-Rel, which show w/o-Att has a similar importance with w/o-Rel.

To further evaluate the robustness of our reward model across responses with varying sub-segment counts, we constructed an additional test dataset. Specifically, we observed that the number of sub-segments in the training set ranged from [4, 16], while the original test set (comprising 500 samples) covered a range of [6, 15]. To assess model performance on longer responses, we collected an additional 200 samples with sub-segment counts between [15, 20]. Our evaluation results indicate an accuracy of 80.4% on the original test set and 79.2% on the newly constructed dataset. The comparable performance across different response lengths suggests that the impact of sub-segment length on model accuracy is minimal, demonstrating the robustness of our model.

During training, a fixed set of hallucination labels is introduced. A key question is whether the method can generalize to new objects that were not present in the training set. To assess the sensitivity of our approach to different object types, we followed prior works (Jiang et al., 2024; Yan et al., 2024) and constructed a dedicated out-of-the-distribution test set based on the Foggy dataset (Cordts et al., 2016). Specifically, we sampled 200 images from the Foggy test set to evaluate the reward model's performance in this setting. The results show an accuracy of 76%, which is lower than the original test set but remains within an acceptable range. This suggests that while the model experiences some performance degradation when encountering unseen object types, it still maintains reasonable robustness.

### 5.4 On Case Study (RQ3)

To get an intuitive understanding of the hallucination mitigation capability of our model, we show two testing results of our method and LLaVA$_{13B}$ in Figure 5. Looking into the generated responses of the first sample, we can learn that by incorporating our fine-grained artificial intelligence feedback, our FGAIF is able

to generate the faithful description for the input visual image, while the baseline cannot (e.g., the baseline generates "A seagull looking out at a lighthouse" and "a boat on the water" mistakenly). This intuitively demonstrates the necessity of considering the fine-grained feedback in reinforcement learning. A similar result can be found in the second sample.

## 5.5 Discussion

To better understand the quality of our AI-based feedback, we manually evaluated the accuracy of the atomic fact labels produced by the model. Specifically, we sampled 90 images from the MSCOCO validation set, generated corresponding answers along with their atomic facts, and manually annotated whether each atomic fact was consistent with the image. This was treated as a binary classification task—hallucination vs. non-hallucination. Based on this evaluation, we found that the accuracy of the AI-based feedback labeling was 85.07%, indicating that while the system is reasonably reliable, there is still room for improvement. These errors highlight the challenge of hallucination detection and suggest that enhancing the accuracy of AI-based feedback remains a promising direction for future work. Improving this component could further boost the effectiveness of our training signal and ultimately improve overall model performance.

## 6 Conclusion

In this paper, we devise an innovative method for refining large vision-language models through Fine-Grained Artificial Intelligence Feedback (FGAIF), which mainly consists of three steps: AI-based feedback collection, fine-grained reward model training, and reinforcement learning with fine-grained rewards. The experimental results on hallucination and general benchmarks show the superiority of our method. The ablation study shows the necessity of each component in our method. In the future, we plan to incorporate more reward models in our method, such as soundness and fluency, which could provide more feedback during the model training stage.

## Acknowledgments

We thank the anonymous reviewers for valuable and insightful feedback. We thank Changchang Sun (Illinois Institute of Technology) and Jialu Li (University of North Carolina, Chapel Hill) for providing editing suggestions on the paper draft. We thank Yushi Hu (University of Washington) and Zhiqing Sun (Carnegie Mellon University) for the discussion on the initial idea. This research is supported in part by the National Science Foundation CAREER Grant IIS-2340435 and an Amazon Research Award. Any opinions, findings, and conclusions or recommendations expressed herein are those of the authors and do not necessarily represent the views, either expressed or implied, of the U.S. Government.

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

## A    Detailed Results

We report the detailed performance on MMHal-Bench and POPE in Table 5 and Table 6.

To understand the performance of our FGAIF, we split the MMHal-Bench into three classes based on the original dataset 1) object existence (class "adversarial object"), 2) object attribute (classes "object attribute" and "counting"), and 3) object relation (class "spatial relation"). From Table 5, we can observe that our method achieves the best performance consistently on all question categories (object existence, object attribute, and object relation), which further demonstrates the effectiveness of our method.

Table 5: Detailed evaluation results for different LMMs on MMHal-Bench. ↓ denotes that the less the value, the better the performance.

| LLM | Overall Score↑ | Hallucination Rate ↓ | Existence | Attribute | Relation |
|---|---|---|---|---|---|
| MiniGPT-4$_{7B}$ | 3.39 | 0.24 | 3.0 | 2.54 | 3.67 |
| mPLUG-Owl$_{7B}$ | 2.49 | 0.43 | 0.33 | 2.58 | 1.5 |
| InstructBLIP$_{7B}$ | 2.10 | 0.58 | 2.08 | 2.67 | 2.17 |
| InstructBLIP$_{13B}$ | 2.14 | 2.75 | 1.75 | 2.82 | 2.5 |
| LLaVA$_{7B}$ | 1.55 | 0.76 | 0.00 | 1.25 | 2.00 |
| LLaVA$_{13B}$ | 1.11 | 0.84 | 0.00 | 1.13 | 1.5 |
| LLaVA-RLHF$_{7B}$ | 2.04 | 0.68 | 1.83 | 2.42 | 2.25 |
| LLaVA-RLHF$_{13B}$ | 2.53 | 0.57 | 2.67 | 2.79 | 2.33 |
| FGAIF$_{7B}$ | **3.09** | **0.36** | **3.58** | **3.21** | **3.33** |

Table 6: POPE evaluation benchmark. Accuracy denotes the accuracy of predictions. "Yes" represents the probability of the model outputting a positive answer. ↑ denotes that the larger the value, the better the performance. The bold font denotes the best performance among our model and baselines with the same backbone model. The underlined font denotes the second-best performance among our model and baselines with the same backbone model.

| Model | Random Acc↑ | F1↑ | Yes | Popular Acc↑ | F1↑ | Yes | Adversarial Acc↑ | F1↑ | Yes | Overall F1↑ | Yes |
|---|---|---|---|---|---|---|---|---|---|---|---|
| MiniGPT-4$_{7B}$ | 79.7 | 80.2 | 52.5 | 69.7 | 73.0 | 62.2 | 65.2 | 70.4 | 67.8 | 74.5 | 60.8 |
| mPLUG-Owl$_{7B}$ | 54.0 | 68.4 | 95.6 | 50.9 | 66.9 | 98.6 | 50.7 | 66.8 | 98.7 | 67.2 | 97.6 |
| InstructBLIP$_{7B}$ | 88.6 | 89.3 | 56.6 | 79.7 | 80.2 | 52.5 | 65.2 | 70.4 | 67.8 | 80.0 | 59.0 |
| InstructBLIP$_{13B}$ | 88.7 | 89.3 | 55.2 | 81.4 | 83.5 | 62.6 | 74.4 | 78.5 | 69.0 | 83.7 | 62.2 |
| LLaVA$_{7B}$ | 50.4 | 66.6 | 98.8 | 49.9 | 66.4 | 99.4 | 49.7 | 66.3 | 99.4 | 66.4 | 99.2 |
| LLaVA$_{13B}$ | 73.7 | 78.8 | 72.3 | 73.6 | 78.2 | 71.0 | 67.2 | 74.4 | 77.8 | 77.1 | 73.7 |
| LLaVA-RLHF$_{7B}$ | 84.8 | 83.3 | 39.6 | 83.3 | 81.8 | 41.8 | 80.7 | 79.5 | 44.0 | 81.5 | 41.8 |
| LLaVA-RLHF$_{13B}$ | 85.2 | 83.5 | 38.4 | 83.9 | 81.8 | 38.0 | **82.3** | 80.5 | 40.5 | 81.9 | 39.0 |
| FGAIF$_{7B}$ | **87.0** | **86.7** | 45.9 | **84.0** | **83.7** | 48.1 | 79.6 | 79.9 | 50.9 | **83.4** | 48.3 |

## B    Error Cases

We show error cases for ChatGPT and LLaVA in Figure 6. In the error case for ChatGPT, ChatGPT should consider the white car as an object attribution rather than object existence. In another case for LLaVA, LLaVA misjudged the consistency between atomic fact and the images.

Error Case for ChatGPT

'The image features a white car parked on the street.
- Entity: White car, street
- Attribute: Color of the car (white),
- Relation: The car is parked on the street.

Error Case for LLaVA

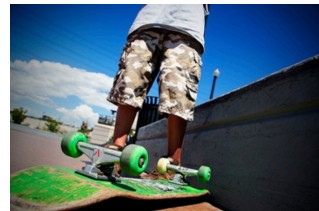

Atomic Fact: A person is standing on a skateboard. LLaVA answer: True

Figure 6: We show error cases for ChatGPT and LLaVA in this figure.

## C  PPO with Fine-Grained Rewards

The algorithm 1 shows in detail how PPO updates the policy LM $P_\theta$ and the value model $V_\psi$, with $K$ fine-grained reward models $\mathcal{R}^{o/a/r}$.

---

**Algorithm 1** Fine-Grained Reinforcement Learning from AI Feedback (FGAIF)

---

**Input:** Initial policy model $P_{\theta_{init}}$; initial value model $V_{\psi_{init}}$; 3 reward models $\mathcal{R}^{o/a/r}$ trained from AI feedback; task prompts $\mathcal{D}$; hyperparameters $\gamma$, $\lambda$, $\epsilon$
**Output:** Updated policy model $P_\theta$.

    Initialize policy model $P_\theta \leftarrow P_{\theta_{init}}$, value model $V_\psi \leftarrow V_{\psi_{init}}$
    **for** step $= 1, \ldots, M$ **do**
        Sample a batch $\mathcal{D}_b$ from $\mathcal{D}$
        Sample output sequence $y^n \sim P_\theta(\cdot \mid x^n)$ for each prompt $x^n \in \mathcal{D}_b$
        Compute rewards $\{r_t^{n,o/a/r}\}_{t=1}^{|y^n|}$ for each sampled output $y^n$ by running $\mathcal{R}^{o/a/r}$
        Compute advantages $\{A_t\}_{t=1}^{|y^n|}$ and value targets $\{V_{\text{targ}}(s_t)\}_{t=1}^{|y^n|}$ for each $y^n$ with $V_\psi$
        **for** PPO iteration $= 1, \ldots, \mu$ **do**
            Update the policy model by maximizing the PPO clipped surrogate objective:

$$\theta \leftarrow \arg\max_\theta \frac{1}{|\mathcal{D}_b|} \sum_{n=1}^{|\mathcal{D}_b|} \frac{1}{|y^n|} \sum_{t=1}^{|y^n|} \min\left( \frac{P_\theta(a_t \mid s_t)}{P_{\theta_{\text{old}}}(a_t \mid s_t)} A_t, \text{clip}(v_t, 1 - \epsilon, 1 + \epsilon) A_t \right)$$

        **end for**
        Update the value model by minimizing a square-error objective:

$$\psi \leftarrow \arg\min_\psi \frac{1}{|\mathcal{D}_b|} \sum_{n=1}^{|\mathcal{D}_b|} \frac{1}{|y^n|} \sum_{t=1}^{|y^n|} \left( V_\psi(s_t) - V_{\text{targ}}(s_t) \right)^2$$

    **end for**

---

## D  Experimental Setups

The version of our ChatPT is gpt-3.5-turbo-0125. We set the number of beams and temperature to 1 and 0 to avoid randomness. The size of the LLaVA 1.5 feedback model is 13B.

# E    Abliation on Hallucination Labeler

To evaluate the dependency of our method on the backbone used for hallucination labeling, we followed your advice and conducted an additional experiment using an LVLM model—InternVL 2.5 (26B)—to label hallucinations. Using the same manually annotated set introduced earlier, we found that InternVL 2.5 achieved an accuracy of 89.47%, which is higher than that of the previously used LLaVA-based hallucination labeler. - Moreover, we re-ran our full pipeline using these improved hallucination labels and observed that better AI-based feedback led to consistently better performance across multiple benchmarks, as shown in Table 7. These results demonstrate that our method generalizes well across different vision-language backbones, and that the overall performance benefits from using stronger hallucination detectors.

| Model | CHAIR_I ↓ | CHAIR_S ↓ | FaithScore ↑ | FaithScore_S ↑ | POPE F1 ↑ |
|---|---|---|---|---|---|
| LLaVA_7B | 9.1 | 22.0 | 89.3 | 71.1 | 66.4 |
| FGAIF_7B | 3.9 | 6.2 | 91.2 | 74.7 | 83.4 |
| FGAIF_7B_new | **3.3** | **5.9** | **91.5** | **75.8** | **84.3** |

Table 7: Comparison of different models using CHAIR, FaithScore, and POPE metrics.

# F    Details of Sentence Splitter

The sub-sentence split process involves two steps: First, we use a spaCy model [2] to segment the input text into coarse-grained sentences. Then, each sentence is further split into sub-sentences using a custom heuristic. In particular, for each sentence splited in the spacy Model, a new sub-sentence is initiated after punctuation marks such as commas, semicolons, exclamation marks, or question marks.

# G    Detailed Discuss

While AI-based feedback significantly reduces human annotation cost, its imperfect accuracy (e.g., 85.07% for LLaVA-13B) introduces potential noise that may affect downstream learning. To understand how such noise propagates through the pipeline, we analyze its impact on two stages: reward model training and reinforcement learning. First, the reward model is trained on AI-generated labels, where label noise may cause incorrect supervision. However, we observe that the model still learns meaningful hallucination annotation, as demonstrated by strong alignment metrics (e.g., 91.2 FaithScore with noisy feedback). This suggests that the reward model exhibits robustness to moderate label noise. Second, during RL training, feedback errors may lead to unstable updates or suboptimal convergence. Empirically, we observe smooth reward curves and no mode collapse, indicating that the reward signal—while imperfect—still provides effective guidance. Overall, while feedback noise is non-negligible, our framework demonstrates empirical stability and effectiveness even with moderately noisy reward signals. Further increasing feedback accuracy (e.g., using InternVL 2.5) yields consistent gains (see Table 7), reinforcing the importance.

To further assess the impact of feedback model quality on the overall performance of our framework, we compared two vision-language models used during the feedback collection stage: LLaVA-13B, achieving 85.07% feedback accuracy, and InternVL 2.5, achieving a higher accuracy of 89.47%. As shown in Table 7, the model trained with InternVL-collected feedback (FGAIF_7B_new) consistently outperforms the LLaVA-based variant (FGAIF_7B) across all evaluation metrics. Specifically, we observed improvements in FaithScore (91.2 → 91.5), FaithScore_S (74.7 → 75.8), and POPE F1 (83.4 → 84.3), along with a reduction in hallucination rates (CHAIR_I: 3.9 → 3.3; CHAIR_S: 6.2 → 5.9). These results highlight that higher-quality feedback leads to better reward modeling and improved fine-tuning outcomes. Importantly, the modular nature of our pipeline allows easy substitution of backbone models, enabling the framework to continuously benefit from advancements in vision-language systems. Additionally, we replaced gpt-3.5-turbo-0125 with gpt-4o-2024-08-06 in our method and found that the performance remained consistent, achieving

---
[2][https://spacy.io/usage/models

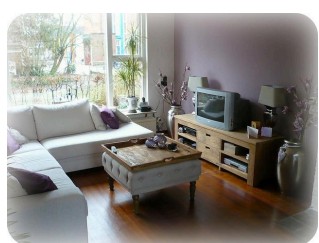

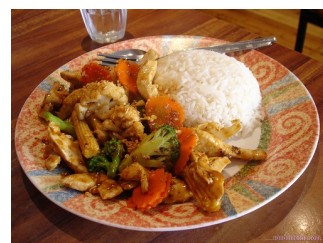

| **Prompt**: Is there a couch in the image? | **Prompt**: Is there a spoon in the image? |
|---|---|
| LLaVA$_{13B}$: Yes.  Ours: Yes. | LLaVA$_{13B}$: Yes.  Ours: Yes. |
| **Prompt**: Is there a chair in the image? | **Prompt**: Is there a bowl in the image? |
| LLaVA$_{13B}$: Yes.  Ours: No. | LLaVA$_{13B}$: Yes.  Ours: No. |

Figure 7: Comparison between the response generated by our method FGAIF and the baseline LLaVA$_{13B}$ on two testing samples. The red fonts denote the generated hallucinations.

an F1 score of 83.5 on the POPE dataset. To further understand the underlying reasons, we sampled 10 data points annotated by both gpt-3.5 and gpt-4o and found a 98% overlap in the generated atomic facts (i.e., the number of overlapping atomic facts divided by the total number of unique atomic facts generated by both models). This indicates that both models produce highly similar atomic facts, leading to comparable performance.

our three-step pipeline—comprising feedback collection, reward model training, and reinforcement learning (RL) fine-tuning—aligns with the standard PPO Reinforcement Learning from Human Feedback (RLHF) framework, which also consists of these three stages. However, our approach introduces a significant enhancement by employing AI-based feedback mechanisms in place of traditional human annotations. This substitution substantially reduces the time and resources typically required for feedback collection, thereby accelerating the overall development process. - In terms of computational costs and training time, our method consists of reward model training and the RL stage, which is same to the other RLHF method [1]. Although our method need to train three reward models and compute rewards with three models in the RL stage, these processes can be run in parallel. Therefore, our approach does not introduce large additional computational overhead beyond the standard framework.

## H  More Qualitative Analysis

Figure 7 presents two additional examples that highlight typical hallucination patterns observed in vision-language models. In the first example, both models correctly identify the presence of a couch, but LLaVA-13B incorrectly predicts the presence of a chair, which is not visible in the image. In the second example, both models correctly identify the spoon, but LLaVA13B hallucinates a bowl, which is absent. These cases reveal a common hallucination tendency: co-occurrence-induced hallucinations. We observe that LLaVA-13B often hallucinates objects that frequently co-occur with the queried item in training data—for instance, assuming a bowl exists when a spoon is detected. This suggests that the model may rely heavily on dataset biases or conditional priors rather than grounding its answers purely in the visual evidence. In contrast, our FGAIF-aligned model correctly suppresses such hallucinations by aligning model behavior with image-grounded atomic facts. Our method explicitly penalizes misalignments during fine-tuning, which improves the model's ability to distinguish what is visually present versus what is statistically likely. This showcases FGAIF's strength in mitigating prior-driven hallucinations and promoting faithful visual understanding.

