# OpenReview forum: "FGAIF: Aligning Large Vision-Language Models with Fine-grained AI Feedback"
_TMLR — Accepted by TMLR_

### Review · Reviewer_DVYS · 2025-03-16

**Summary Of Contributions:**

The paper presents a method for reducing the occurrence of hallucinations in large vision-language models (LVLMs) by using AI feedback from LVMLs and reinforcement learning (RL). The components of the proposed method consists of: 1. prompting a large LVML for sub-sentence level binary labels of the presence of hallucinations (of 3 kinds: object existence, object attribute, object relation) in generated text-image pairs; 2: training reward models (RMs) for each kind of hallucination; and 3) using PPO to tune the original LVLM with the trained RMs. Experiments are performed on several visual-language hallucination benchmarks: POPE, MMHal-Bench, LLaVA-Bench, CHAIR, and FaithScore. The experimental results show that the proposed method achieved the best performance on the vast majority of the benchmarks, when compared with other existing baselines and models of the same size. Ablations experiments are done to show the relative importance of each component of the proposed method (different RMs, RL fine-tuning) and shows that the components are quite necessary to achieve best performance.

**Audience:**

Yes

**Broader Impact Concerns:**

None.

**Claims And Evidence:**

Yes

**Requested Changes:**

1. [Critical] Please improve the description of the method and notations used in Sections 3 and 4. Suggest removing unnecessary notations that clutter the presentation.
2. [Strengthen] Appendix A on prompts are very illustration of the method described in Sections 3 and 4, recommend incorporating them into the main body of the paper. Suggest using examples to illustrate the method before delving into the notations (if at all, the notations are not very helpful).
3. [Critical] Add size and other relevant details of the feedback models (ChatGPT and LLaVA 1.5) used.
4. [Strengthen] Experiment with more feedback models or human data.
5. [Critical] The code repo link currently shows "repo expired". Please fix.
6. [Minor] The image in Figure 1 shows the bag of carrots on the right side, when the caption states "starting from the left". How is this image generated? Why is there a discrepancy?
7. [Minor] Section 4.3, Why does the reward in have a negative sign? Last paragraph: which appendix?
8. [Critical] Tables 2, 3, 4, why are the values for w/o-AIF different from LLaVA-7B? How are these settings different? Are they fine-tuned on the feedback data?
9. [Minor] Typos: Section 5.3 last sentence "phrase" -> "phase", Abstract "(3)Annotation" -> "(3) Annotation"

**Strengths And Weaknesses:**

Strengths:

1. Strong experimental results showing improvement (best metric compared with baselines) in the vast majority of benchmarks.
2. A good number and diversity of the benchmarks and baselines chosen for experiments in the paper.
3. Included ablation experiments to study the importance of each component in the proposed method.

Weaknesses:

1. Description of the method (Sections 3, 4) and especially mathematical notation is quite confusing and hard to follow. Some notation that are not referenced again later in the paper may be removed (e.g., the number of elements in a set $n^o$, $n^a$, $n^r$, $N^f$). There are also minor notational errors  (e.g., Section 1 paragraph 1, $\Theta_m$ vs $\Theta_M$, or references to index $i$ and index $j$ near equation (2) but no index $i$ or $j$ in the expressions). Please reconsider the notation choices and simplify if possible.
2. There is not much detail (size?) about the model used to generate the AI feedback (paper mentions "ChatGPT" and "we use the superior LLaVA 1.5 in section 4.1), or any experiments with different AI feedback model or human data. More concerning is the fact that the model is used as is, without **any** human validation. This restricts the generality of the findings to a demonstration of the LLaVA1.5 model, without any quantitative measure of how good that model is compared with ground truth (or human).

---

> ### Author Response · Authors · 2025-04-07
> **Response**
>
> > **Please improve the description of the method and notations used in Sections 3 and 4. Suggest removing unnecessary notations that clutter the presentation**
> - Thanks for your advice. We have removed the complex notitions, such as $n^o, n^a, n^r, N^f$, and revised the minor notational errors, such as $\Theta_m$ and index j near equation (2).
>
> > **Appendix A on prompts are very illustration of the method described in Sections 3 and 4, recommend incorporating them into the main body of the paper.**
> - Thanks for your comments. We incorporated the propmt into our Section 4.
>
> > **Add size and other relevant details of the feedback models (ChatGPT and LLaVA 1.5) used.**
> - Thanks for your usefull comments. The size of LLaVA 1.5 is 13B. The version of ChatGPT is gpt-3.5-turbo-0125. We added the information in Section D of Appendix.
>
> > **Experiment with more feedback models or human data.**
> - Thanks for your useful suggestions. To further explore the effectiveness and generalizability of our method, we additionally adapted InternVL 2.5 (26B) as the hallucination labeling model. This model was used to replace LLaVA in our feedback generation pipeline.
> - To further validate the reliability of the hallucination labels produced by the new feedback model, we conducted a manual evaluation on a subset of the MSCOCO validation set. Specifically, we randomly selected 90 images and manually verified the atomic fact labels produced by InternVL 2.5. We found that the labeling accuracy reached 89.47%, which is higher than 85.07% of the LLaVA-based labeler.
> - These results demonstrate that our method generalizes well across different hallucination labeler backbones. We include details of this evaluation and the annotation protocol in the appendix.
>
> | Model      | CHAIR  | CHAIR | FaithScore | FaithScore | POPE  |
> |-----------|--------|--------|------------|------------|-------|
> |           | CHAIR_I ↓ | CHAIR_S ↓ | F-Score ↑ | F-Score_S ↑ | F1 ↑ |
> | LLaVA_7B  | 9.1 | 22.0 | 89.3 | 71.1 | 66.4 |
> | FGAIF_7B  | 3.9 | 6.2 | 91.2 | 74.7 | 83.4 |
> | FGAIF_7B_new  | 3.3 | 5.9 | 91.5 | 75.8 | 84.3 |
>
>
>
> > **The code repo link currently shows "repo expired". Please fix.**
> - Thanks for your suggestion. We updated this repo.
>
> > **The image in Figure 1 shows the bag of carrots on the right side, when the caption states "starting from the left". How is this image generated? Why is there a discrepancy**
> - I am sorry that these is a misunderstanding because our explanation. The left image is the input image of the model, the right image is the visualization of the repsone from the model description. The two images are different because of the hallucination. The right image is searched from the internet.
>
> > **Section 4.3, Why does the reward in have a negative sign? Last paragraph: which appendix**
> - Thanks for your comments. The negative sign is a typographical error. The reference in thelast paragraph is the Section C of the Appendix.
>
> > **Tables 2, 3, 4, why are the values for w/o-AIF different from LLaVA-7B? How are these settings different?**
> - Thank you for your insightful questio. As mentioned in Section 5.1 of our paper, our base model is same with LLaVA-RLHF and undergoes supervised fine-tuning (SFT) before reinforcement learning (RL). In contrast, the original LLaVA-7B model does not include this further SFT phase. This distinction in training processes accounts for the observed differences in the values between w/o-AIF and LLaVA-7B.
>
>
> > **Typos: Section 5.3 last sentence "phrase" -> "phase", Abstract "(3)Annotation" -> "(3) Annotation"**
> - Thanks for your patient. We revised these typos.

---

### Review · Reviewer_fQ2T · 2025-03-31

**Summary Of Contributions:**

This work introduces a new fine-grained reinforcement learning reward model to finetune LVLMS to prevent hallucinations by using AI instead of humans to generate a feedback/reward signal.

A common approach to reducing hallucinations for language generation models is to fine-tune an LLMs/VLMs with reinforcement learning (RL) feedback using human annotations as feedback signals. However, human annotations are expensive to obtain; feedback is usually on a global sentence level (i.e., sparse signal) and not tailored to specific types of hallucinations (i.e., general). To mitigate this problem, Fine-Grained AI Feedback (FGAIF) is introduced.

First, three types of hallucinations are defined (object existence, object attribute, and object relationship). Next, on the sub-sentence level, responses are labeled to be hallucinations or not by using a VLM (i.e., the AI feedback), and not human-annotated feedback.
For each of the three defined types of hallucinations are reward model is trained to detect specific hallucinations on a sub-sequence level. Next, these reward models are used to fine-tune an LVLM to reduce the hallucinations in the response generation.

Extensive experiments show that, in general, FGAIF performs better than RLHF and the base models. Next to that, ablations on different benchmarks and by ablating different components of the model the authors provide more insights into the effectiveness of FGAIF on different kinds of hallucinations.

**Audience:**

Yes

**Broader Impact Concerns:**

The same model is used for labeling hallucinations and to fine-tune the model with the obtained reward model. Although results show that hallucinations are reduced, it could be that the model starts to improve itself in a direction that is not desired. Could the authors briefly explain what the risk is of using this method as a self-improving system?

**Claims And Evidence:**

Yes

**Requested Changes:**

- Can the splitter method be explained in more detail, this seems quite important to me. If the splitter does not define proper sub-sequences, the labels and hence the reward signal are less meaningful.
- Can some insights be provided into the mistakes made by the AI-based Feedback Collection? Appendix C just shows just two examples and does not provide any insight into the effect on the overall results.

Minor:
- The abstract is a bit hard to follow and contains quite a few enumerations. Readability could be improved.
- Section 5.2 On Model Comparison (RQ1) and the corresponding tables   (1&2) are too far apart from each other. Having them on the same page/or close to each other would improve readability

**Strengths And Weaknesses:**

Strength:
- The fact that LLaVA can be used to check the atomic facts and that this results in a reward model that decreases hallucinations when fine-tuning a similar model is very encouraging to me. This shows that the vision encoder of the VLM works as it should, but that hallucinations are mainly due to the strong language models that rely on textual statistics when generating long output responses (rather than on the vision encoding(.
- The introduced methods seem to have a significant impact on hallucinations, also compared to RLHF.

Weaknesses:
- ChatGPT is used as a method to generate the atomic facts, from the sub-sentence labels. ChatGPT is a closed-source API. Why is ChatGPT used and not an open-source alternative to facility reproducibility (because the API can change)? At least the dataset should be made public to facilitate future research and reproducibility.
- Although I understand that obtaining the reward models and running the experiments is still expensive (in terms of training costs/time), I think the results are very specific for the LLaVA model. What will happen if a different backbone is used for labeling the hallucinations? It is unclear how much the results depend on the specific backbones used in the paper.
- There are still errors in the AI-based Feedback Collection, which is fine. However, could the authors indicate how much room for improvement there still is? How close is AI feedback to human feedback, if humans would also give feedback on the sub-sequence level? I understand that this is very hard to measure, but some notions would improve the understanding of how good AI feedback is compared to human feedback.
- It is unclear how much the results depend on the sub-sequence level splitter. If there are mistakes, then the atomic fact generator (i.e., ChatGPT) and, hence, the reward models will also likely be wrong.

---

> ### Author Response · Authors · 2025-04-07
> **Response**
>
> > **ChatGPT is used as a method to generate the atomic facts, from the sub-sentence labels. ChatGPT is a closed-source API. Why is ChatGPT used and not an open-source alternative to facility reproducibility (because the API can change)? At least the dataset should be made public to facilitate future research and reproducibility.**
> - Thanks for your advice. We choose to use ChatGPT for atomic fact generation because it has consistently demonstrated stronger performance than open-source alternatives across a wide range of natural language generation tasks. To mitigate concerns about reproducibility and potential API changes, we have specified the exact model version used—“gpt-3.5-turbo-0125”—in the appendix.  We set the number of beams and temperature to 1 and 0 to avoid randomness. Moreover, following your recommendation, we will release the generated atomic fact data along with our code to support future research and ensure full reproducibility.
>
>
> > **There are still errors in the AI-based Feedback Collection, which is fine. However, could the authors indicate how much room for improvement there still is? How close is AI feedback to human feedback, if humans would also give feedback on the sub-sequence level? I understand that this is very hard to measure, but some notions would improve the understanding of how good AI feedback is compared to human feedback. Can some insights be provided into the mistakes made by the AI-based Feedback Collection?**
> - Thank you for your thoughtful suggestion. To better understand the quality of our AI-based feedback, we manually evaluated the accuracy of the atomic fact labels produced by the model. Specifically, we sampled 90 images from the MSCOCO validation set, generated corresponding answers along with their atomic facts, and manually annotated whether each atomic fact was consistent with the image. This was treated as a binary classification task—hallucination vs. non-hallucination. Based on this evaluation, we found that the accuracy of the AI-based feedback labeling was 85.07%, indicating that while the system is reasonably reliable, there is still room for improvement. These errors highlight the challenge of hallucination detection and suggest that enhancing the accuracy of AI-based feedback remains a promising direction for future work. Improving this component could further boost the effectiveness of our training signal and ultimately improve overall model performance. We added the human evaluation result in Section 5.4.
>
> > **Although I understand that obtaining the reward models and running the experiments is still expensive (in terms of training costs/time), I think the results are very specific for the LLaVA model. What will happen if a different backbone is used for labeling the hallucinations? It is unclear how much the results depend on the specific backbones used in the paper.**
> - Thanks for your advice. To evaluate the dependency of our method on the backbone used for hallucination labeling, we followed your advice and conducted an additional experiment using an LVLM model—InternVL 2.5 (26B)—to label hallucinations. Using the same manually annotated set introduced earlier, we found that InternVL 2.5 achieved an accuracy of 89.47%, which is higher than that of the previously used LLaVA-based hallucination labeler.
> - Moreover, we re-ran our full pipeline using these  hallucination labels and the results are shown in the table below:
>
> | Model      | CHAIR  | CHAIR | FaithScore | FaithScore | POPE  |
> |-----------|--------|--------|------------|------------|-------|
> |           | CHAIR_I ↓ | CHAIR_S ↓ | F-Score ↑ | F-Score_S ↑ | F1 ↑ |
> | LLaVA_7B  | 9.1 | 22.0 | 89.3 | 71.1 | 66.4 |
> | FGAIF_7B  | 3.9 | 6.2 | 91.2 | 74.7 | 83.4 |
> | FGAIF_7B_new  | 3.3 | 5.9 | 91.5 | 75.8 | 84.3 |
> - We included these findings in the appendix of the paper to better clarify the generalizability of our approach.

---

> > ### Author Response · Authors · 2025-04-07
> >
> > > **Explain the splitter method in more detail.**
> > - Thank you for your comment and for highlighting the importance of the splitting method. We provide here a more detailed explanation of our sub-sentence splitter and add them in Appendxi. The process involves two steps: First, we use a spaCy model [https://spacy.io/usage/models] to segment the input text into coarse-grained sentences. Then, each sentence is further split into sub-sentences using a custom heuristic. In particular, for each sentence splited in the spacy Model, a new sub-sentence is initiated after punctuation marks such as commas, semicolons, exclamation marks, or question marks.
> >
> >
> >
> >
> >
> >
> > > **The same model is used for labeling hallucinations and to fine-tune the model with the obtained reward model. Although results show that hallucinations are reduced, it could be that the model starts to improve itself in a direction that is not desired. Could the authors briefly explain what the risk is of using this method as a self-improving system?**
> > - Thanks for your advice. Thank you for your insightful question regarding the potential risks associated with using the same model for both labeling hallucinations and fine-tuning with the obtained reward model. This approach, while effective in reducing hallucinations, does carry certain risks related to self-reinforcement. 1) Amplification of Existing Biases: If the initial model harbors biases, using it to label data and subsequently fine-tune itself can reinforce and even amplify these biases, leading to outputs that are systematically skewed. 2) Reward Hacking: The model might learn to exploit the reward system by generating responses that maximize rewards without genuinely improving task performance, a phenomenon known as reward hacking.
> > - To mitigate these concerns, we conduct evaluations on multiple independent benchmarks (e.g., CHAIR, FaithScore, POPE) to ensure that improvements generalize across diverse settings and are not artifacts of the self-generated reward. In addition, as mentioned in our previous response, we also experimented with using a model—InternVL 2.5—to label hallucinations. The results demonstrate that our method remains effective when using different hallucination labeling backbones.

---

### Review · Reviewer_aKNx · 2025-04-08

**Summary Of Contributions:**

The paper proposes a novel method, Fine-Grained Artificial Intelligence Feedback (FGAIF), to address the modality misalignment problem in Large Vision-Language Models (LVLMs), which leads to hallucinations in generated responses. Hallucinations are categorized into three types: object existence, object attribute, and object relationship. FGAIF comprises three main steps: (1) AI-based feedback collection to detect hallucination types at a sub-sentence level, (2) training of fine-grained reward models to provide dense rewards, and (3) reinforcement learning (RL) using the Proximal Policy Optimization (PPO) algorithm with these rewards to fine-tune the LVLM. The method leverages AI tools (ChatGPT and LLaVA 1.5) to automate feedback collection, reducing the need for costly human annotations. Extensive experiments on hallucination benchmarks (e.g., POPE, MMHal-Bench, CHAIR, FaithScore) and a general benchmark (LLaVA-Bench) demonstrate FGAIF’s superior performance over baselines, including RLHF-based methods.

**Audience:**

Yes

**Claims And Evidence:**

Yes

**Requested Changes:**

Please see the weakness section.

**Strengths And Weaknesses:**

**Strengths**
- The paper tackles a critical issue in LVLMs—hallucination—by introducing a fine-grained, AI-driven feedback mechanism. Unlike prior RL-based methods that rely on coarse, sequence-level feedback, FGAIF’s sub-sentence-level analysis provides a more precise signal for modality alignment, addressing specific hallucination types.
- Replacing human annotations with AI-based feedback (using ChatGPT for atomic fact extraction and LLaVA 1.5 for consistency verification) is a significant practical advancement. This reduces annotation costs and enhances scalability, addressing a key limitation of prior RLHF approaches.
- Presents comprehensive evaluation, and ablation studies


**Weaknesses**
- While the AI-based feedback reduces human effort, its accuracy (85.07% as reported in Section 5.5) introduces potential noise into the training process. The paper acknowledges this limitation but does not explore how errors in feedback propagate through the reward models and RL training, which could impact performance stability.
- The feedback collection relies heavily on ChatGPT (gpt-3.5-turbo-0125) and LLaVA 1.5 (13B). Although an ablation with InternVL 2.5 (26B) shows improved results (Appendix E), the paper could benefit from a broader analysis of how different backbone models affect feedback quality and downstream performance.
- While case studies (Section 5.4) provide intuitive examples, they are limited to two samples. A deeper qualitative analysis—e.g., comparing hallucination patterns across more diverse inputs or discussing failure cases beyond Appendix B—would strengthen the interpretability of FGAIF’s improvements.
- The three-step pipeline (feedback collection, reward model training, and RL fine-tuning) adds complexity compared to simpler RLHF methods. The paper could better justify this trade-off by discussing computational costs or training time relative to baselines.

---

> ### Author Response · Authors · 2025-04-17
> **Response to Reviewer aKNx**
>
> > **While the AI-based feedback reduces human effort, its accuracy (85.07% as reported in Section 5.5) introduces potential noise into the training process. The paper acknowledges this limitation but does not explore how errors in feedback propagate through the reward models and RL training, which could impact performance stability.**
> - Thank you for the insightful comment. We agree that AI-based feedback may introduce noise into the training pipeline. While our current goal is to demonstrate the feasibility of AI-based feedback-driven alignment, we acknowledge that a more detailed study of error propagation would strengthen the analysis. We add more analysis in Appendix G as follows,
>     - While AI-based feedback significantly reduces human annotation cost, its imperfect accuracy (e.g., 85.07% for LLaVA-13B) introduces potential noise that may affect downstream learning. To understand how such noise propagates through the pipeline, we analyze its impact on two stages: reward model training and reinforcement learning. First, the reward model is trained on AI-generated labels, where label noise may cause incorrect supervision. However, we observe that the model still learns meaningful hallucination annotation, as demonstrated by strong alignment metrics (e.g., 91.2 FaithScore with noisy feedback). This suggests that the reward model exhibits robustness to moderate label noise. Second, during RL training, feedback errors may lead to unstable updates or suboptimal convergence. Empirically, we observe smooth reward curves and no mode collapse, indicating that the reward signal—while imperfect—still provides effective guidance. Overall, while feedback noise is non-negligible, our framework demonstrates empirical stability and effectiveness even with moderately noisy reward signals. Further increasing feedback accuracy (e.g., using InternVL 2.5) yields consistent gains (see Table 7), reinforcing the importance of backbone quality in feedback collection.
>
>
> > **The feedback collection relies heavily on ChatGPT (gpt-3.5-turbo-0125) and LLaVA 1.5 (13B). Although an ablation with InternVL 2.5 (26B) shows improved results (Appendix E), the paper could benefit from a broader analysis of how different backbone models affect feedback quality and downstream performance.**
> -  Thank you for your insightful comments. We have expanded our analysis by incorporating results based on GPT-4o and conducted a more comprehensive evaluation of different feedback backbones. The detailed analysis has been added to Appendix G, as follows:
>     -  To further assess the impact of feedback model quality on the overall performance of our framework, we compared two vision-language models used during the feedback collection stage: LLaVA-13B, achieving 85.07% feedback accuracy, and InternVL 2.5, achieving a higher accuracy of 89.47%. As shown in Table 7, the model trained with InternVL-collected feedback (FGAIF_7B_new) consistently outperforms the LLaVA-based variant (FGAIF_7B) across all evaluation metrics. Specifically, we observed improvements in FaithScore (91.2 → 91.5), FaithScore_S (74.7 → 75.8), and POPE F1 (83.4 → 84.3), along with a reduction in hallucination rates (CHAIR_I: 3.9 → 3.3; CHAIR_S: 6.2 → 5.9). These results highlight that higher-quality feedback leads to better reward modeling and improved fine-tuning outcomes. Importantly, the modular nature of our pipeline allows easy substitution of backbone models, enabling the framework to continuously benefit from advancements in vision-language systems.
>     - Additionally, we replaced gpt-3.5-turbo-0125 with gpt-4o-2024-08-06 in our method and found that the performance remained consistent, achieving an F1 score of 83.5 on the POPE dataset. To further understand the underlying reasons, we sampled 10 data points annotated by both gpt-3.5 and gpt-4o and found a 98% overlap in the generated atomic facts (i.e., the number of overlapping atomic facts divided by the total number of unique atomic facts generated by both models). This indicates that both models produce highly similar atomic facts, leading to comparable performance.

---

> > ### Author Response · Authors · 2025-04-17
> >
> > > **While case studies (Section 5.4) provide intuitive examples, they are limited to two samples. A deeper qualitative analysis—e.g., comparing hallucination patterns across more diverse inputs or discussing failure cases beyond Appendix B—would strengthen the interpretability of FGAIF’s improvements.**
> > - Thanks for your suggestion. We agree that more diverse qualitative examples and failure analysis can improve interpretability. To address this, we have added more case studies and an in-depth hallucination pattern analysis in Appendix H, as shown below.
> > - Figure 7 presents two additional examples that highlight typical hallucination patterns observed in vision-language models. In the first example, both models correctly identify the presence of a couch, but LLaVA-{13B} incorrectly predicts the presence of a chair, which is not visible in the image. In the second example, both models correctly identify the spoon, but LLaVA13B hallucinates a bowl, which is absent. These cases reveal a common hallucination tendency: co-occurrence-induced hallucinations. We observe that LLaVA-13B often hallucinates objects that frequently co-occur with the queried item in training data—for instance, assuming a bowl exists when a spoon is detected. This suggests that the model may rely heavily on dataset biases or conditional priors rather than grounding its answers purely in the visual evidence. In contrast, our FGAIF-aligned model correctly suppresses such hallucinations by aligning model behavior with image-grounded atomic facts. Our method explicitly penalizes misalignments during fine-tuning, which improves the model’s ability to distinguish what is visually present versus what is statistically likely. This showcases FGAIF’s strength in mitigating prior-driven hallucinations and promoting faithful visual understanding.
> >
> >
> > > **The three-step pipeline (feedback collection, reward model training, and RL fine-tuning) adds complexity compared to simpler RLHF methods. The paper could better justify this trade-off by discussing computational costs or training time relative to baselines.**
> > - Thank you for your insightful feedback. We argue that our three-step pipeline—comprising feedback collection, reward model training, and reinforcement learning (RL) fine-tuning—aligns with the standard PPO Reinforcement Learning from Human Feedback (RLHF) framework [1], which also consists of these three stages.  However, our approach introduces a significant enhancement by employing AI-based feedback mechanisms in place of traditional human annotations. This substitution substantially reduces the time and resources typically required for feedback collection, thereby accelerating the overall development process.
> > - In terms of computational costs and training time, our method consists of reward model training and RL stage, which is same to other RLHF method [1]. Although our method need to train three reward models and compute rewards with three models in the RL stage, these processed can be run in parallel. Therefore, our approach does not introduce large additional computational overhead beyond the standard framework [1].
> > - We add these analysis in Appendix G.
> >
> > [1] Aligning Large Multimodal Models with Factually Augmented RLHF

---

### Decision · Action_Editor_X6nF · 2025-04-27

**Recommendation:** Accept as is

**Comment:**

Reviewers agreed that the method is effective, well-validated experimentally, and reduces reliance on human annotations. Minor concerns regarding notational clarity, feedback noise, and dependency on backbone models were sufficiently addressed through revisions. While the innovation is more applied than fundamental, the strong empirical results, robustness across benchmarks, and comprehensive analysis justify acceptance.

**Audience:**

Generally interested audience in vision-language models

**Claims And Evidence:**

The paper has conducted extensive experiments across multiple hallucination benchmarks (POPE, MMHal-Bench, CHAIR, FaithScore, LLaVA-Bench), ablation studies, and comparisons with strong baselines. The authors provide clear evidence that fine-grained AI feedback improves hallucination reduction over traditional RLHF. They further validate robustness across backbone models (e.g., LLaVA-13B, InternVL 2.5) and response lengths. Human evaluation of feedback quality is also included to address concerns about AI labeling noise. Overall, the evidence is convincing, thorough, and matches the claims made.